# A Comparative Analysis of Anticardiolipin, Anti-Β2-Glycoprotein-1, and Lupus Anticoagulants in Saudi Women with Recurrent Spontaneous Abortions

**DOI:** 10.3390/jpm13010002

**Published:** 2022-12-20

**Authors:** Osama Abdulrahman Shaikhomar, Syed Tabrez Ali

**Affiliations:** Department of Physiology, Faculty of Medicine, Umm Al-Qura University, Makkah P.O. Box 715, Saudi Arabia

**Keywords:** anticardiolipin, glycoprotein, lupus anticoagulant, recurrent spontaneous abortion, Saudi women

## Abstract

Association and pathogenesis of antiphospholipid antibodies (APAs) in patients with Antiphospholipid syndrome (APS) as well as systemic lupus erythematosus (SLE) suffering from unexplained spontaneous abortions is controversial. Moreover, the majority of the published evidence is based on the patient histories rather than the study groups. The main objective of the present study is to do a comparative analysis of the high levels of antiphospholipid antibodies (APAs) including anticardiolipin (ACA), anti-β2-glycoprotein-1 (β2 GP1), and Lupus anticoagulants (LA) and their association with the gestational age of abortion and duration past abortion in Saudi women. In this study, 100 women living in Makkah city, located in the Western region of Saudi Arabia, with a mean age of 30.5 ± 12.60 years (mean ± standard deviation), having a previous history of recurrent spontaneous abortions were enrolled as case group and 100 healthy pregnant women previously having one or more successful pregnancies were selected as control group. Based on the gestational age of the patient’s miscarriages, our results revealed consistently and significantly high levels of ACA, β2 GP1, and LA, being greatest at more than 15 weeks when compared with 10 weeks and 11–15 weeks, respectively. Based on duration of past miscarriages, a slightly increased level was observed at ≤4 years after their first miscarriage. In addition, less or more equal levels of these antibodies were observed at 5–12 and more than 12 years in all the cases of miscarriages. We concluded an elevated pattern of APAs in these patients with an increase in the age of abortion. A comparison between the levels of ACA, β2 GP1, and LA further indicated a highly significant level of LA in all the cases of abortions (*p* < 0.0005).

## 1. Introduction

Antiphospholipid syndrome (APS) generally refers to pregnancy complications including pre-eclampsia, placental malfunctioning, retarded fetal growth, and often repeated miscarriages [1,2]. The main cause of these complications is reported to be due to anticardiolipin antibodies (ACAs) present in the circulating blood of these patients [3].

ACAs lead to thrombosis after binding with phospholipids by inhibiting the release of gonadotropins as well as placental anticoagulant proteins [4,5]. Although the exact mode of action of the placental associated thrombosis of these antibodies is not clear, it is suggested in the literature that these antibodies may cause local acute inflammatory responses and neutrophil infiltration, which ultimately ends up with fetal loss [6].

Animal studies have shown spontaneous fetal loss due to passive transfer of purified anticardiolipin IgG [7]; however, because of the normalization in the level of these antibodies during pregnancy, not only are the chances of miscarriages reduced, but an improvement in the survival rates of these fetuses has also been found [8].

Although around 0.2% to 2% of pregnant women are affected with (APS) in the general population, the chances of recurring miscarriages are reported to be about 10% in these patients [9,10]. The main cause of this situation is believed to be due to the presence of these antiphospholipid antibodies (APAs) belonging to a class of acquired heterogeneous autoantibodies directed against negatively charged phospholipids and phospholipid-binding proteins in these patients [11,12].

Moreover, along with ACAs, lupus anticoagulant (LA) has also shown obstetric complications in 15–20% of patients with a relatively higher rate of recurrent miscarriages, about 50–75% due to severe pre-eclampsia and unexplained intrauterine fetal death [13]. It is thus clear in clinical practice that ACAs and LA are the two most frequently used APAs antibodies. During the experimental trials to characterize these antibodies, many new possible target antigens like β2 GP1, prothrombin, annexin V, protein C, and protein S, have also been discovered [14,15,16], thus suggesting that rather than phospholipid, phospholipid binding proteins and/or phospholipid-protein complexes actually contribute to the actual target of APAs.

It is now established that high levels of APAs are directly associated with spontaneous miscarriage along with thrombocytopenia and multiple thrombosis [17]. However, the clinical mode of action and the pathogenesis of APS are still not clear, and the prevention and treatment of repeated spontaneous abortion associated with acute promyelocytic leukemia (APL) still remains controversial.

Regarding APAs associated pregnancy loss; our literature survey indicates very limited studies with the Saudi Arabian population. The current study has therefore been designed to determine and compare an association between Anticardiolipin, Anti-Β2-Glycoprotein-1, and Lupus Anticoagulants in Saudi women with recurrent spontaneous abortions.

## 2. Materials and Methods

### 2.1. Study Population and Design

The current study was conducted on Saudi women living in Makkah city, located in the western region of Saudi Arabia. In accordance with the Declaration of Helsinki, after getting the ethical approval from the affiliated Institutional Review Board (IRB) committee at Umm Al-Qura University, Makkah, Saudi Arabia [HAPO-02-K-012-2019-11-245] and a written informed consent indicating the purpose of the study, benefits, risks, and their rights to withdraw from the study, women having a mean age of 30.5 ± 12.60 years were enrolled in the study. Prior to the study, a complete physical checkup of all the candidates was carried out by a registered physician. In the first step of the screening, women with a previous history of any digestive, renal, endocrine, rheumatic, or nutritional disease including malnutrition (weight loss, low body mass index or special dietary habits), undergoing a surgical process, blood donation or blood transfusion within six months, strenuous exercise or heavy manual labor were excluded. Similarly, candidates with hypertension (systolic pressure ≥150 mmHg and/or diastolic pressure ≥95 mmHg), or heavy smokers (smoking >20 cigarettes/day) were also excluded from the study. In the second step of screening, candidates with positive Hepatitis B surface antigen, Hepatitis C antibodies, or HIV antibodies were also excluded from the study. Finally, 100 Saudi women with a history of two or more abortions without any history of drug abuse or medical illness were selected as the case group. (In the case group, history including parity, live birth, number of miscarriages, gestational age at each miscarriage, and duration past miscarriage was determined).

Abortion is defined as either fetal loss around the gestational age of 15 weeks or the weight of fetus below 500 g. For the control group, 100 healthy women of the same age group without any history of miscarriages and previously having one or more successful pregnancies were selected.

### 2.2. Data Collection

In a clinical facility, by vein puncture, 5 milliliters of blood was drawn and allowed to clot for 30 min at room temperature. After centrifugation for 10 min at 3400 r.p.m (ALC centrifuge PK130 Missouri City, TX, USA), serum was collected in the sterile tubes and the samples were analyzed for ACA, and anti-β2-glycoprotein-1 (β_2_ GP1) antibodies.

ACA level was measured by an enzyme linked immunoassay test using Orgentec kits (ORGENTEC Diagnostika, Mainz, Germany). Standardization and quantitation of results are based on the work of Loizou et al. [18], using the polyclonal ‘Harris’ standards [19]. Anti β2-GP1 antibodies were assayed with commercial ELISA kits (QUANTA Lite^TM^ β_2_ GP1, INOVA Diagnostics Inc., San Diego, CA, USA) for semi-quantitative determination. The cutoff value was determined to be 8 U/mL as defined by the manufacturer.

As described previously [20,21], LA was assayed in fresh plasma samples using validated in-house lupus ratio (LR) tests. Two LR tests were performed, one based on the activated partial thromboplastin time (LR-APTT) and the other based on the Russell viper venom time (LR-RVVT). The LR tests were performed on a 1:1 mixture of patient plasma and pooled normal plasma. For each of the LR tests, two coagulation times were measured, one with a reagent with a low phospholipid concentration and the other with a high phospholipid concentration. The ratio between the two coagulation times (low phospholipid/high phospholipid concentration) was divided by the corresponding ratio obtained with pooled normal plasma. According to the method described previously, the final ratio is defined as the LR of that patient’s plasma [21].

### 2.3. Data Analysis

Statistical analysis of the collected data was done by Student *t tests* using SPSS program 17.0 (SPSS Institute, Inc.; Chicago, IL, USA) and the relation between mean of antiphospholipid antibodies and gestational age at the time of miscarriage (≤10 week, 11–15 weeks, >15 week) and duration past abortions (≤4 year, 5–l2 years, >12 years) were determined. Results were tabulated as mean ± standard deviation (SD). In all the cases, a *p* value < 0.05 was considered as statistically significant.

## 3. Results

Demographic and clinical data for the case and control groups is presented in Table 1.

All patients were Saudi women while pregnancy morbidity referred to at least one fetal loss around 10 weeks of pregnancy.

As shown in Figure 1 and Figure 2 respectively, a slightly high level of ACA, Anti â2-GP1 and LA was observed in the case group when compared with the control group (*p* < 0.05).

Data for the average distribution of ACA, Anti â2 GP1 and LA, based on gestational age of miscarriages, is presented in Figure 3A–C, respectively. At the age of around 10 weeks of miscarriage (Figure 3A), an increased level of ACA, Anti â2 -GP1 and LA were observed (*p* < 0.005). Based on gestational age of miscarriages, 11–15 weeks (Figure 3B), and more than 15 weeks (Figure 3C), a highly significant increase in the levels of ACA, Anti â2 GP1 and LA was observed, being highest at the gestational age more than 15 weeks of miscarriage (*p* < 0.0005). Based on duration past miscarriages as shown in Figure 4A, a slightly increased level was observed at ≤ 4 years after their first miscarriage. In addition, less or more equal levels of these antibodies were observed at 5–12 and more than 12 years in all the cases of miscarriages (Figure 4B,C, respectively).

These results thus indicate a greater rate of mean antiphospholipid antibodies with increasing gestational age at the time of abortion, which remained at a high level in the first four years, and later antibodies began to fall.

## 4. Discussion

Association and pathogenesis of ACAs in patients with systemic lupus erythematosus as well as in the patients suffering from unexplained spontaneous abortions is controversial. Although a weak correlation of adverse effects of ACAs has been found in a large obstetric group [22], Love et al. [23] could not find any such adverse correlation of ACAs in the patients with a history of fetal miscarriage in non-systemic lupus erythematosus (SEL). In a previously published study, in comparison with 993 controls, the prevalence of ACAs was reported in 331 women with a history of at least one spontaneous abortion or fetal death [24], thus suggesting that ACAs may not be considered as a risk factor for fetal loss in the absence of a prior adverse obstetric event. In contrast, a direct relationship between polyclonal and monoclonal ACAs and recurrent miscarriage has been derived from patients with antiphospholipid syndrome [25]. In another investigation, in a group of 108 women suffering from placental infractions, about 4% showed a positive association of anticardiolipin antibodies [26]. However, no such direct relationship between ACAs and spontaneous miscarriage has been reported either for in vivo animal or for in vitro human studies [27]. Another investigation showed the involvement of both anticardiolipin and anti-phosphatidyl serine antibodies for the recurrent abortion in these women; however, the chief cause was reported to be ACAs [28].

Although much knowledge has been acquired regarding the clinical manifestations of antiphospholipid syndrome, the role of antiphospholipid antibodies in the pathogenesis of this disorder remains poorly understood.

A literature survey indicates that Sapporo classification criteria for antiphospholipid syndrome, first proposed in 1999 [2], updated at the Eleventh International Congress on Antiphospholipid Antibodies in Sydney in 2006 [29] are most often used in practice as diagnostic criteria for these antibodies. We have therefore interpreted our results as defined by the above mentioned preliminary or modified Sapporo criteria.

In our results, we found a consistent increase in the level of ACA with the duration of gestational age of abortions. Whenever the gestational age was older in all the abortions, ACA was always found to be higher. Furthermore, in all the cases, a slightly increased level of ACA was observed in the first 4 years past all abortions but decreased after more than 5 years. These results are in conformity with the previous findings where a major cause of recurrent fetal loss was found to be anticardiolipin antibodies [30].

It is interesting to note that, although 2006 Sydney antiphospholipid syndrome classification criteria has included IgG/IgM anti-cardiolipin, IgG/IgM anti-β2-glycoprotein antibodies and lupus anticoagulant, some patients suffering from antiphospholid syndrome still remain sero-negative (sero-negative APS or SNAPS). These patients are always at a risk of recurrent pregnancy loss or arterial/venous thrombosis, since “non-critical” antiphosphlipid antibodies are detected in these patients, which may produce unusual types of thrombosis, or some kind of unexplained early or late recurrent abortions, especially in young women [31]. Screening and diagnosis of these SNAPS patients is therefore highly recommended to prevent any obstetric or thrombotic adverse effect. Furthermore, it is reported that, in the general population, apart from ACAs, other antiphospholipid antibodies are also involved in causing recurrent abortion. For example, in the cases of antiphospholipid syndrome with or without autoimmune disorders, anti-annexin V, anti-prothrombin, and anti-β-Sub2-glycoprotien-1, antibodies seem to play an important role in recurrent pregnancy fetal loss [32,33]. In particular, anti- β2 GP1 antibodies- β2 GP1 complexes seem to have a more positive role in the activation of complement cascade. β2 GP1, the most well recognized main target of antiphospholipid antibodies, is widely represented on trophoblast and decidual surface [34]. Moreover, elevated levels of lupus anticoagulant and anticardiolipin, which are known as the two best characterized antiphospholipid antibodies, are also reported to be closely involved with recurrent pregnancy loss [35,36]. These findings are in close conformity with our results, where the percentage of LA was found to be significantly higher (*p* < 0.0005) than ACA and β2 GP1 in all the studied groups.

In the clinical trials, a strong correlation between the number of abortions and the high level of LA and ACAs has been reported [37,38,39], which is again in agreement with our study, where we found a positive correlation between high levels of ACA, β2 GP1 and LA with an increase in the age of gestational miscarriage.

Since the differences in a number of physiological and environmental factors including the dosage, timing, and frequency of treatment and outcome assessment parameters may produce differences in the results, it is highly recommended to investigate the immuno-pathological mechanism, which is involved in the fetal loss, as well as the mode of action of the therapeutic agents used for the treatment of such cases.

Most recently, modified protein C-activated thrombin generation assay (TGA) has been reported in the literature, which determines the thrombogenicity of these antiphospholipid antibodies of all groups of patients, especially those who are at a higher risk of antiphospholipid complications [40]. Therefore, it seems highly beneficial to evaluate these antibodies in combination with modified TGA test for better understanding and clarification of the association of recurrent spontaneous abortions and these antiphospholipid antibodies.

## 5. Conclusions

Based on the results of the current study, it is concluded that elevated levels of APAs are the main cause of spontaneous recurrent abortions in Saudi women. A comparative analysis of these antibodies further showed similar elevated levels for ACA and β2 GP1, but LA in all the cases showed highly significant elevated level, thus suggesting that these APAs assays are an effective diagnostic tool for the detection of recurrent spontaneous abortions.

## Figures and Tables

**Figure 1 jpm-13-00002-f001:**
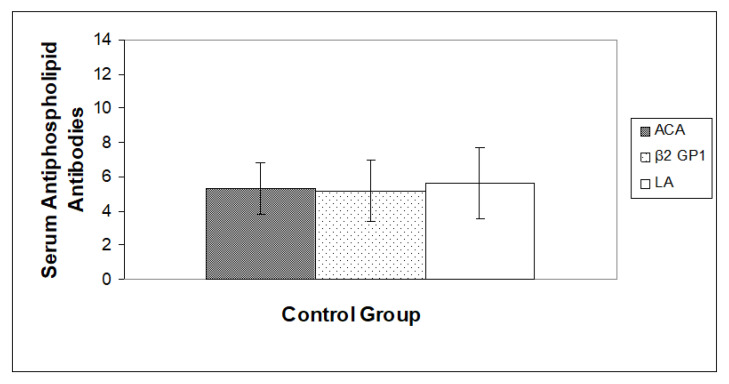
Estimation of anticardiolipin (ACA), anti-β2-glycoprotein-1 (β2 GP1) and Lupus anticoagulants (LA) in the control group of Saudi women. Values are Mean ± SD, (*n* = 100), *n* = Total number of subject examinations.

**Figure 2 jpm-13-00002-f002:**
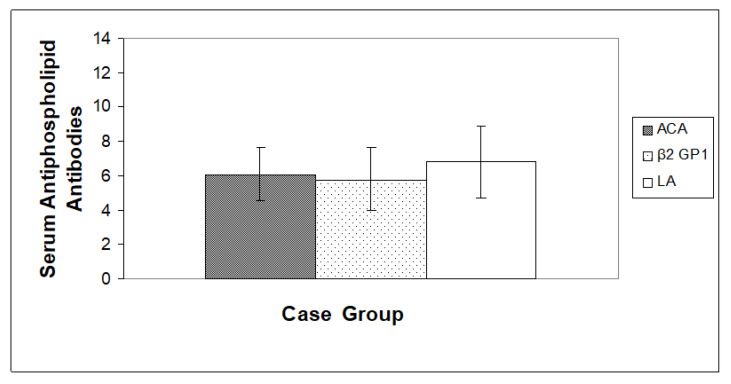
Estimation of anticardiolipin (ACA), anti-β2-glycoprotein-1 (β2 GP1) and Lupus anticoagulants (LA) in the case group of Saudi women. Values are Mean ± SD, (*n* = 100), *n* = Total number of subject examinations.

**Figure 3 jpm-13-00002-f003:**
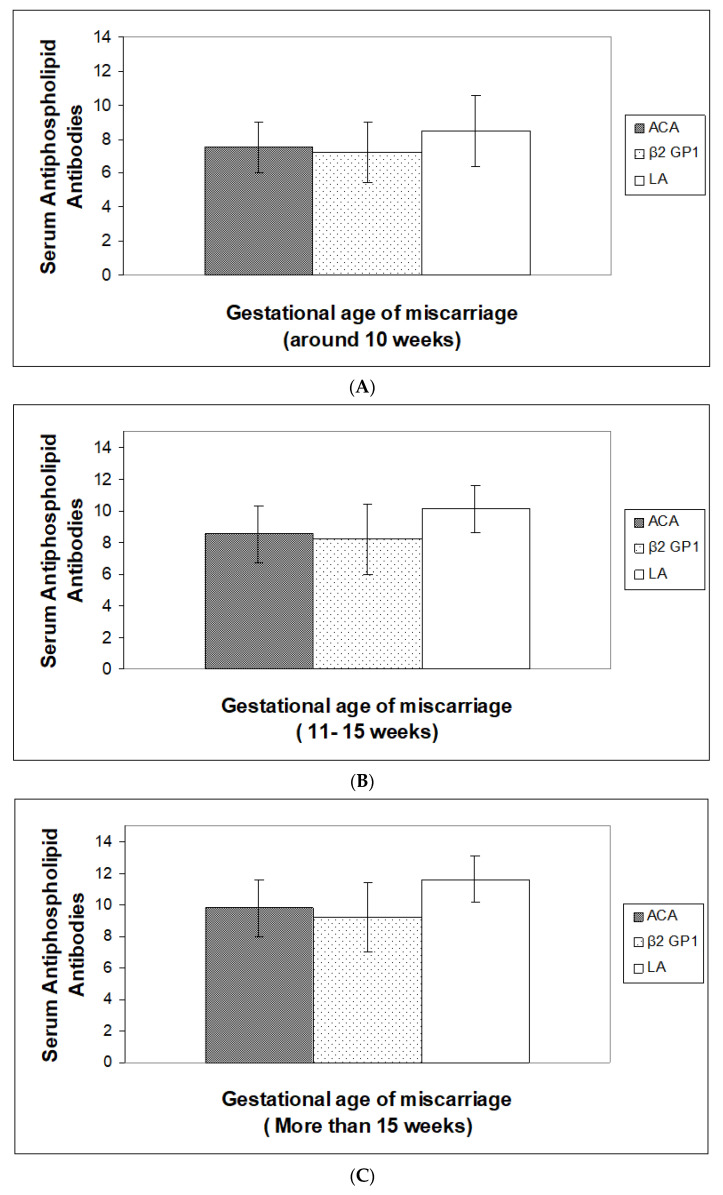
(**A**) Estimation of anticardiolipin (ACA), anti-β2-glycoprotein-1 (β2 GP1) and Lupus anticoagulants (LA) based on gestational age of miscarriages (≤10 weeks) in Saudi women. Values are Mean ± SD, (*n* = 100), *n* = Total number of subject examinations. (**B**) Estimation of anticardiolipin (ACA), anti-β2-glycoprotein-1 (β2 GP1) and Lupus anticoagulants (LA) based on gestational age of miscarriages (11–15 weeks) in Saudi women. Values are Mean ± SD, (*n* = 100), *n* = Total number of subject examinations. (**C**) Estimation of anticardiolipin (ACA), anti-β2-glycoprotein-1 (β2 GP1) and Lupus anticoagulants (LA) based on gestational age of miscarriages (>15 weeks) in Saudi women. Values are Mean ± SD, (*n* = 100), *n* = Total number of subject examinations.

**Figure 4 jpm-13-00002-f004:**
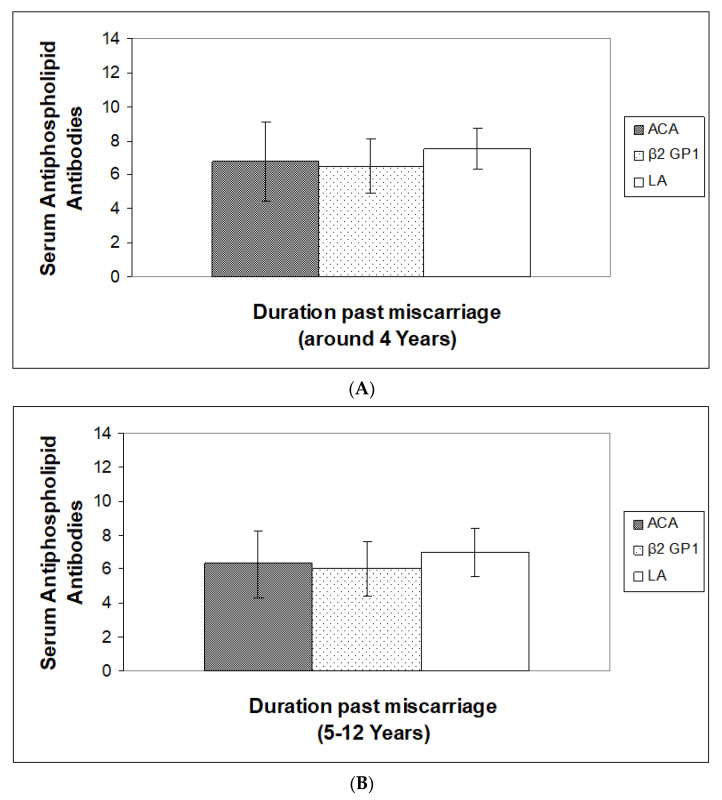
(**A**) Estimation of anticardiolipin (ACA), anti-β2-glycoprotein-1 (β2 GP1) and Lupus anticoagulants (LA) based on duration past miscarriages (≤4 years) in Saudi women. Values are Mean ± SD, (*n* = 100), *n* = Total number of subject examinations. (**B**) Estimation of anticardiolipin (ACA), anti-β2-glycoprotein-1 (β2 GP1) and Lupus anticoagulants (LA) based on duration past miscarriages (5–12 years) in Saudi women. Values are Mean ± SD, (*n* = 100), *n* = Total number of subject examinations. (**C**) Estimation of anticardiolipin (ACA), anti-β2-glycoprotein-1 (β2 GP1) and Lupus anticoagulants (LA) based on duration past miscarriages (>12 years) in Saudi women. Values are Mean ± SD, (*n* = 100), *n* = Total number of subject examinations.

**Table 1 jpm-13-00002-t001:** Demographic and clinical data of study populations.

Characteristics	Case Group	Control Group
Number of Patients (*n*)	100	100
Sex (female)	100	100
Age ± SD; years old	30.5 ± 12.60	30.5 ± 12.60
Race (Saudi): Caucasian/Black	81/19	80/20
Body Mass Index (BMI)	28.52 ± 6.81	28.08 ± 6.52
Thrombosis, Y/N	88/12	0/100
Systemic Lupus Erythematosus (SLE), Y/N	77/23	26/74
Diabetes, Y/N	0/100	1/99
Placental Abruption, Y/N	0/100	2/98
Placenta Previa, Y/N	0/100	2/98
Inherited Thrombophilia * Y/N	2/98	1/99
Pregnancy Loss, Y/N	96/04	0/100

* Inherited thrombophilia: factor V Leiden, prothrombin G20210A polymorphism, antithrombin, protein C or protein S deficiency.

## Data Availability

Not applicable.

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
