# Peer review of "A Comparative Analysis of Anticardiolipin, Anti-Β2-Glycoprotein-1, and Lupus Anticoagulants in Saudi Women with Recurrent Spontaneous Abortions"

_jpm, 2022, doi:10.3390/jpm13010002_

Round 1
Reviewer 1 Report
This study represents confirmation of the known relationship between antiphospholipid antibodies and pregnancy morbidity. The only novelty observed is the the analysis of the specific cohort.
My major concern regarding this study is the presentation of the results observed. Authors shoul include tables with the demographic data for both groups analyzed and variables analyzed.
In the Discussion section, authors should inform us if the diagnosis of antphospholipid syndrome has been established according to valid Sidney criteria (2006). Probably, the consultation with the specialist in this area could be helpful. Even though the study has been conducted in the Physiology department, the clinical issue has been addressed and it has the impact on everyday clinical practice (bearing in mind that this study confirmed what is already known).
Author Response
Dear Dr.
With reference to Manuscript ID: jpm-2036746 (Type of manuscript: Article), Title: A Comparative Analysis of Anticardiolipin, Anti-Β2-Glycoprotein-1 and Lupus Anticoagulants in Saudi Women with Recurrent Spontaneous Abortions, Authors: Osama Abdulrahman Shaikhomar *, Syed Tabrez Ali, please find point by point details regarding the referees comments.
Review Report Form ONE
- As mentioned by the reviewer, demographic data for both groups analyzed and variables analyzed are added in the revised manuscript (page no: 19)
- In the Discussion section, diagnosis of antphospholipid syndrome according to valid Sidney criteria (2006) has been described as asked by the referee (page no: 9).

Reviewer 2 Report
Dear autors,
I have a few questions
essential
1 :can you explain the measured values of beta-2GPI, ACA and lupus anticoagulans - the relation to calibrators (harris standards for ACA ??), the range of measured values and the calculation of the result, especially for LA.
2. Can you add a cut-off for the monoclonal methods, it is quite essential for any further conclusions drawn
3. Can you use the cut-off to describe how many patients were positive in your cohort.
4. Can you substantiate the claim that antibodies are the most common cause of miscarriage - I assume recurrent.
minor
Abstract
5- Can you explain that the sample consisted of women aged 24-40 years with a mean of 30.5 +- 12.6 , which is 27.9 - 43.1
6- In the introduction it would be good to mention what does your prenatal screening look like, which may invalidate the data on miscarriage.
7. On page 4 line 79 you mention severe antibody positivity associated with thrombocytopenia - did you have such a patient? Was/wasn't she ruled out by t-test?
8 The data collection section erroneously also contains an incomplete description of the methodologies which is significant - please add. As well as the addition of LA testing from citrate plasma, but that is a detail.
I will comment on the results after clarification of the previous comments
Discussion
It is interesting however outdated. In recent years, according to EndNote, at least 15 - 20 papers per year have been published on the topic, which would be worth discussing.
Translated with www.DeepL.com/Translator (free version)
Author Response
Dear Dr.
With reference to Manuscript ID: jpm-2036746 (Type of manuscript: Article), Title: A Comparative Analysis of Anticardiolipin, Anti-Β2-Glycoprotein-1 and Lupus Anticoagulants in Saudi Women with Recurrent Spontaneous Abortions, Authors: Osama Abdulrahman Shaikhomar *, Syed Tabrez Ali, please find point by point details regarding the referees comments.
Review Report Form TWO
Essential questions
- Measured values of beta-2GPI, ACA and lupus anticoagulans - the relation to calibrators (harris standards for ACA) is as follows:
Anti-Cardiolipin calibration was done by Harris sera (IRP 97/656 (IgG), HCAL (IgG) / EY2C9 (IgM). Quantitative calculation was done with the range of 0-90 U/ml, Cut-off 10 U /ml, coating (cardiolipin, Cofactor: beta -2-GPI.
Anti-beta-2-Glycoprotein calibration was done by Harris sera (IRP 97/656 (IgG), HCAL (IgG) / EY2C9 (IgM). Quantitative calculation was done with the range of 0-90 U/ml, Cut-off 10 U /ml, coating (beta-2-glycoprotein I)
Regarding LA, the ratio between the two clotting times was divided by the corresponding ratio for the normal pooled plasma. This final ratio is the lupus ratio of that plasma. The overall sensitivity was found to be 95.1%. When the results were grouped in low, medium and high positive plasmas, a “consensus” regarding the strength of each plasma was found to be 85.0% of the results.
Cut-off value was determined using guidelines of International Society on Thrombosis and Haemostasis Scientific and Standardization Subcommittee (ISTH-SSC, 2009) that recommends application of the 99th percentile to determine the cutoff value. [Pengo V, Tripodi A, Reber G, et al. Subcommittee on Lupus Anticoagulant/Antiphospholipid Antibody of the Scientific and Standardisation Committee of the International Society on Thrombosis and Haemostasis. Update of the guidelines for lupus anticoagulant detection. J Thromb Haemost. 2009; 7(10): 1737- 1740.]
- Cut-off for the monoclonal methods has been added as asked by the referee (page no:6)
- According to cut-off around 96% patients were positive in the case group.
- Substantiate the claim that antibodies are the most common cause of miscarriage - I assume recurrent, comes from the analysis of the results which indicate a consistently and significantly high levels of ACA, β2 GP1, and LA, being greatest at more than 15 weeks when compared with 10 weeks and 11-15 weeks, respectively. In addition, at 5-12 and more than 12 years past miscarriage in the case group, level of these antibodies returned back to almost control level.
Minor questions
- In the abstract, mean age of the women has been corrected (page no: 2)
- On page 4 line 79 regarding severe antibody positivity associated with thrombocytopenia, no such patient was added.
- Data collection section has been improved with the description of the methodologies highlighted in red color.
- Discussion section has been updated with most recent information.

Round 2
Reviewer 1 Report
Table 1 should be rearranged. Please, include percentages of variables studied and include p values so that readers can be informed that both groups are comparable. Please, correct the names of groups studied in the pdf form (both groups are assigned as 'case group' instead of case and controls. It seems that a significantly higher percentage of females with SLE were included in the case group. Could that be the explanation for higher levels of antiphospholipid antibodies? The authors should discuss this finding.
Besides discussion regarding present antiphospholipid syndrome diagnostic criteria, authors should inform us if the diagnosis of the antiphospholipid syndrome has been confirmed in the case group (how many of them and if they were treated? how?)
Author Response
All corrections are made in Blue color.
See the main manuscript file.
- Percentages of variables studied and p values are included in Table-1
- Correction is done in the Table-1 as Case group and Control Group.
- Significantly higher percentage of females with SLE included in the case group could be an explanation for higher levels of antiphospholipid antibodies but probably not the exact cause of spontaneous recurrent abortion. A literature survey has indicated the prevalence of a clinically significant antiphospholipid antibodies profile in SLE patients to be about 20%. Lupus anticoagulant test positivity and antiphospholipid antibodies positivity correlates better with antiphospholipid antibodies-related clinical events. However, it is not always necessary that a positive antiphospholipid antibodies test in the SLE patients is clinically significant having a risk of antiphosholipid related clinical manifestations particularly pregnancy loss. As reported by Ozan etal, no increase in the risk of fetal loss was observed in 96 SLE patients with 132 pregnancies with the diagnosis of antiphospholipid antibodies and /or antiphosholipid syndrome. [Ozan, U. Stephane, Z. Doruk, E. The clinical significance of antiphospholipid antibodies in systemic erythematosus. Eur. J. Rheumatol. 2016, 3(2), 75-84].
In addition, although increased lupus activity in women suffering from SLE has been reported to causes preeclampsia and sometimes still birth, however, in most of the cases, successful pregnancies are reported in these women [Megan, EB. Clowse, MD. Lupus Activity in Pregnancy. Rheum. Dis. Clin North Am. 2007, 33(2), 237–v]. We thus conclude that increased level of antiphospholipid antibodies rather than SLE itself may be the cause of recurrent abortions in our case group.

Reviewer 2 Report
Dear Authors, thank you for answering my questions. However, there is still considerable uncertainty that you understand the results you have measured. The number of 96% positive patients is unusually high and is related to a very low cut-off. Please consider dividing the evaluated data in all groups into positive and negative and for those, process the average values or better the median. Furthermore, please consider evaluating the number of positive patients in all groups according to the clinical cut-off and adapting the evaluation of the conclusions accordingly. The conclusions should include the effect of the level on the clinical phenomenon - in this case repeated misscariage.
Author Response
All corrections are made in BLUE color.
See the main manuscript file.
- Comments and Suggestions are highly appreciated. Just to clarify that the value 96 % is the pregnancy loss, not positive patients.
As shown in Table-1, a significantly higher percentage of females with SLE included in the case group could be an explanation for higher levels of antiphospholipid antibodies but probably not the exact cause of spontaneous recurrent abortion. A literature survey has indicated the prevalence of a clinically significant antiphospholipid antibodies profile in SLE patients to be about 20%. Lupus anticoagulant test positivity and antiphospholipid antibodies positivity correlates better with antiphospholipid antibodies-related clinical events. However, it is not always necessary that a positive antiphospholipid antibodies test in the SLE patients is clinically significant having a risk of antiphosholipid related clinical manifestations, particularly pregnancy loss. As reported by Ozan etal [000] no increase in the risk of fetal loss was observed in 96 SLE patients with 132 pregnancies with the diagnosis of antiphospholipid antibodies and /or antiphosholipid syndrome. [Ozan, U.; Stephane, Z.; Doruk, E. The clinical significance of antiphospholipid antibodies in systemic erythematosus. Eur. J. Rheumatol. 2016, 3(2), 75-84.] In addition, although increased lupus activity in women suffering from SLE has been reported to causes preeclampsia and sometimes still birth, however, in most of the cases, successful pregnancies are reported in these women [000]. [Megan, EB; Clowse, MD. Lupus Activity in Pregnancy. Rheum. Dis. Clin North Am. 2007, 33(2), 237–v].
We thus conclude that increased level of antiphospholipid antibodies rather than SLE itself, may be the cause of recurrent abortions
